# Microclimate Classification of Bologna (Italy) as a Support Tool for Urban Services and Regeneration

**DOI:** 10.3390/ijerph18094898

**Published:** 2021-05-04

**Authors:** Marianna Nardino, Letizia Cremonini, Teodoro Georgiadis, Emanuele Mandanici, Gabriele Bitelli

**Affiliations:** 1CNR-IBE (National Research Council, Institute for BioEconomy), 40129 Bologna, Italy; marianna.nardino@ibe.cnr.it (M.N.); letizia.cremonini@ibe.cnr.it (L.C.); 2Department of Civil, Chemical, Environmental, and Materials Engineering, University of Bologna, 40136 Bologna, Italy; emanuele.mandanici@unibo.it (E.M.); gabriele.bitelli@unibo.it (G.B.)

**Keywords:** urban planning, urban morphology, urban sustainability, population wellness and health

## Abstract

A microclimate classification of the entire Bologna Municipality has been carried out in order to give a tool to the local administration in the drafting of the General Urbanistic Plan (PUG). The city was classified considering the variation of air temperature as a function of the surface characteristics, the vegetation fraction, the building density and the H/W ratio (height to width). Starting from the microclimate analysis carried out with fluid-dynamic modeling (Envi-met) for some areas of the city of urban interest, the air temperature variation was correlated to the physiological equivalent temperature (PET) in order to make a classification of physiological well-being for the resident population. An urban map of a normalized microclimate well-being index (BMN) has been obtained to give support when private, and public actors want to regenerate part of the city, taking into account the climate-centered approach for the development of a sustainability city.

## 1. Introduction

To make cities and human settlements inclusive, safe, resilient and sustainable—these are the main aims of the goal n#11 of the Sustainable Development Goals (SDGs) of the UN agenda [1,2]. In 2008, for the first time in history, the global urban population outnumbered the rural population, and by 2050, it is expected that two-thirds of the world population will be living in urban areas [3]. Even in the last 30 years, the population living in slums decreased from about 46% to 23%, the exposure to life dangers and to health condition detriment is very high. The main impacts are expected from increased air pollution, the occurrence of extreme weather events, and the thermal regime of cities [4]. Because of these possible and expected risks, resilience is assumed by policymakers as a new paradigm in which the city should be pro-active in favoring the inclusion where all citizens are equally protected in terms of dignity and health [5].

On 15 October 2015, the Covenant of Mayors [6] set new and more ambitious goals and broadened its range of action by becoming the Covenant of Mayors for Climate and Energy. It means that not only climate change mitigation has to be taken into account but also adaptation, a fact that strongly boosts municipalities toward new concepts related to the urban regeneration projects, utilizing as a new tool the adaptation plans to rethink the city in terms of fragilities, strategies, and actions to build a new sustainable and resilient place to live in, where the main threat for the very next future is the climate change [7,8,9]. The Covenant of Majors, along with SDGs, are fundamental milestones to boost the regeneration of cities.

The adaptation plans (AP’s) can serve as primary guidelines to depict the current fragilities, to design strategies, and to plan actions to solve the fragilities. As far as the cities and climate are concerned, an AP strongly interacts with the structure of the urban fabric. Materials, location and density of buildings and the presence of vegetation are parameters, which couple the occurrence of heatwaves to the urban heat island in determining the physiological wellness conditions of the population [5,10,11].

Different parts of a city respond in different ways in terms of thermal regimes, thus impacting citizens [12]. There is a clear need to know existing microclimate conditions of the urban texture to proceed to apply strategies and to take actions. This strongly reflects on the urban policy such as the town building regulation which is the primary tool to address a real sustainable and resilient development of a city. 

Therefore, knowledge of the current state is an indispensable precondition for planning the development of the city and securing the historically consolidated parts. 

The Municipality of Bologna, the first cities in Italy to have adopted itself with an adaptation plan according to the new Covenant of Mayors (BLUEAP, Bologna Adaptation Plan for a Resilient City) [13], understood the need for adaptation actions to be effectively reported in an urban regulation plan. The actions, based on climate themes, became an integral part of urban development and not linked to sporadic and unplanned interventions [14]. 

Facing the expected climate changes which affect cities and in order to preserve the cultural heritage, the way of life, and the citizen’s health, the Municipality decided to provide the city with a new urban plan. The law provides that the new PUG put together the three urban planning tools that are in force today: the Municipal Structural Plan, the Urban Building Regulations and the Municipal Operational Plans, and comply with the new contents established by the Emilia-Romagna Region (Regional Law 24/2017) [15]. It is, therefore, a substantial change and a transition between two different models of local government. The PUG adopted represents the synthesis of a path that acknowledges that the urban expansion is over, and it is necessary to work by regenerating the existing city and the urban community that inhabits it. To obtain the planned results, three significant objectives have been identified:Resilience and environment;Habitability and inclusion;Attractiveness and work.

Each of the three objectives defines four urban-scale strategies that introduce essential changes in a planning strategy. Thus, the strategies regarding “resilience and the environment” are concerned with ensuring health and well-being for those who live in the city today and for those who will inhabit it tomorrow. The focus is on minimizing the risks of climate change and supporting the energy transition and the circular economy.

The Bologna Municipality, relying on the existence of an extensive database together with the historical development of the city, decided to improve the urban knowledge with a microclimate analysis by developing an instrument that identifies an index of fragility. This index relies on the existence, in the urban structure, of all those elements that contribute to regulating the bioclimatic conditions of interest for the resident population. 

It is well known that the urban form and its variation strongly influence the thermal patterns of the city [16,17,18,19]: for this reason, the technical staff of the Planning Office of the Municipality of Bologna started from the “Territory Classification” map which characterizes and identifies urban and building textures and morphology similar to each other within the municipal area [20]. The city’s climate analysis is computed, taking into account the morphological indices together with vegetation cover and land surface temperature giving support to classify the entire urban environment in climatic classes.

The main purpose of this new methodological approach is to provide the technical experts of the municipalities with a tool directly developed to be able to use the databases normally held by the municipalities for urban planning activities. This is necessary due to the widespread scarcity of human resources to be dedicated specifically to complex modeling studies, and therefore favoring in this way a prompt response to the requests of the territory.

## 2. Materials and Methods

### 2.1. The Municipality of Bologna

Capital of the Metropolitan city and of the Emilia Romagna Region (Italy), Bologna (445,075 N; 113,514 E) is located between the mountains of the Tuscan-Emilian Apennines and the heart of the Po Valley (Figure 1). A meeting point between North and South, between East and West, the Adriatic Riviera as well as Venice, Florence, Milan and Rome can be easily reached from the city. Bologna has a warm and temperate climate, and there is significant rainfall throughout the Metropolitan territory. According to Köppen and Geiger the climate has been classified as Cfa [21]. The average annual temperature in Bologna is 14.0 °C, and the average annual rainfall is 774 mm (ARPAe Emilia Romagna, Idro-MeteoClima Service) [22]. The urban resident population is about 400,000 citizens.

The city of Bologna presents a particular dichotomy in its architecture: its city center is built over ancient structures of the Roman times and is mainly characterized by low-rise buildings dating back to the medieval and renaissance eras, with relatively narrow roads and a constant presence of arcades. The suburbs are instead much more recent with an increased presence of high-rise buildings and larger roads. The road system is characterized by a radial organization which converges toward the city center, and the latter is surrounded by a ring road dating back to the 20th century. Bologna’s city center and its arcades are still a lively area for the citizens of Bologna: most of Bologna’s historical shops (active for at least 50 years, as stated by the regional law of 10 March 2008 n.5) are located in this area [23].

The historical urban structure of Bologna is characterized by relatively concentric circles (Figure 2). The oldest walls, of which remain visible today only a few remains, are those of the so-called “Selenite circle”, built in correspondence with the sunset of the Western Roman Empire. The expansion of the city and the birth of new villages outside the walls gave rise to the need to build a new circle of walls, and this is called the circle of a thousand, Cerchia dei Mille. This second wall was about 3.5 km long and had 18 doors, also called Menageries or Torresotti (Figure 3), all surmounted by a tower and now all demolished except four still visible. Outside the second wall, the city is characterized by another circle, corresponding to the so-called garden city, full of liberty villas, tree-lined streets, and private vegetation. All the main public gardens in Bologna are placed along with the garden city where public and private vegetation find the fullest synthesis. The garden city, therefore, finds its continuity solution to the North in the Railway Station which, being a polar center of Italian rail transport, represented over time a factor of significant economic development [24].

### 2.2. Microclimate Analysis

The microclimate analysis was performed through the three-dimensional non-hydrostatic microclimate model ENVI-met [27]. This model is designed to simulate the surface-plant-air interactions within the daily cycle in an urban environment with a typical resolution of 0.5 to 10 m in space and 10 s in time. Several variables can be simulated, included flow around and between buildings, exchange processes of heat and vapor between ground, wall and vegetation surfaces and the atmosphere, bioclimatology and particle dispersion [28]. The model was validated as reported in Yang et al. [29,30].

In order to run the model, the user must have detailed data on soil characteristics, buildings, vegetation and initial and boundary atmospheric conditions for the area of interest. The file used to spatial input information has a user-friendly graphical interface. The correct use and set-up of the model for a specific area requires a background in meteorology and a good knowledge of the study area domains to be simulated. ENVI-met can be used for several studies to test various urban canyon aspects as well as ratios and orientation effects on outdoor thermal comfort, the role of vegetation in the mitigation of the urban heat island effect, and other factors.

In agreement with the Urban Office of the Bologna Municipality, five areas of 1 Km × 1 Km were chosen in order to have an entirely urban area characterization for the whole Bologna city (Figure 4). 

With respect to the areas under study, Bolognina, Masi and Corticella areas can be considered as falling into a high level of climatic vulnerability. Roveri and Barca areas were included in the study to have a complete mapping of the urban buildings types present in the municipal area, allowing the analysis of an industrial and productivity area (Roveri).

Each 1 km^2^ size area was simulated with 5 m cell resolution (200 × 200 cells) and with the same meteorological initial conditions. The vertical resolution was set to 2 m (25 cells) to reach a vertical height of 50 m and the lowest gridbox was split into five subcells. It was decided to simulate the micro-climate conditions for the day with the maximum temperature recorded during the 2017 summer heatwave. To this end, an analysis of the meteorological data was carried out, obtaining the free data ‘Dexter System’ of the Idro-MeteoClima Service of the ARPAe, Emilia Romagna Region [31]. The urban weather station of Bologna was chosen which is located in the center of the city and therefore represents an urban condition: the day that recorded the maximum temperature in 2017 was 4 August.

The hourly data were then downloaded from the ARPAe weather station and entered into the model for its initialization and for boundary conditions:Wind speed: 2.4 m/s;Wind direction: 220° from North;Maximum air temperature: 39.6 °C at 14:00 p.m.;Minimum air temperature: 27.9 °C at 5:00 a.m.;Maximum relative humidity: 40% at 02:00 a.m.;Minimum relative humidity: 17% at 12:00 p.m.

### 2.3. Morphological Climate Classes

The urbanized territory of Bologna has been characterized according to Morphological Climate classes (MC) considering that the physical variables that define the climate in an urban environment are:Surface temperature: the map of the surface temperatures at 90 m resolution, obtained from satellite data (ASTER) during a heatwave (acquired image on 23 June 2017 at 12:10 a.m.). was available for the entire municipal area, defining the type of surface and the degree of heating due to it [32,33] (Figure 5);Fraction of Vegetation: percentage of vegetation coverage for the entire Municipality of the city of Bologna derived from Sentinel-2 satellite data (Figure 6);Building density: it is a morphological parameter defined as the fraction between the volume of the building and the surface that is occupied. The building density was computed by adding the volumes of buildings belonging to a block divided by the total area of the block (Figure 7);H/W ratio: another morphological index usually used to define an urban canyon given by the ratio between the average height of the buildings and the width of the road. The calculation of the H/W ratio (building height/road width) required approximations as each block was surrounded by several streets: through an automated procedure, the ratio between the average of the heights of the buildings of each block was calculated divided by the average of the widths of the roads that surrounded it (Figure 8).

The data available from Bologna Municipality were already organized on a “block” scale (defined as a block of buildings surrounded by four streets) therefore we decided to keep this as a surface base, calculating and defining the maps and indexes on this basis of GIS map.

The surface temperature and vegetation cover for the entire Bologna Municipality are reported in Figure 5 and Figure 6.

The results of these four input maps are shown in Figure 5, Figure 6, Figure 7, Figure 8. From these maps, it can be immediately seen how the surface temperature is higher in the city’s historical center and in other areas present in the urban fabric. The normalized difference vegetation index is obtained by the red and near-infrared bands of Sentinel-2 multispectral imagery through the operation (band_NIR − band_RED/band_NIR + band_RED). The index clearly shows the areas with no vegetation. The morphological indexes describe the areas with denser areas of buildings and geometries where the H/W ratio is most remarkable.

Once the maps with the input data for the calculation of the morphological climate classes had been defined, every single variable has been related to the variation of the air temperature.

The following equations were used, obtained from the literature:Ta = 0.373Ts + 17.691(1)
∆Ta = −0.34VF(2)
∆Ta = 0.39 BD(3)
∆Ta = 7.54 + 3.97ln(H/W)(4)
where ∆Ta is the variation in air temperature in °C [34], Ts is the surface temperature obtained from the satellite in °C, VF is the fraction of vegetation [35], and BD is the building density [35] and Equation (4) is obtained by [36]. 

Many studies were made to compute the relationship between air temperature and surface temperature from satellite [37,38,39]: the uncertainties are significant, especially in urban environments, the topic of current scientific interest [40,41,42].

The authors are keen to emphasize that this methodology does not want to represent the atmospheric conditions of an urban area in terms of turbulent exchanges, radiative balance and air temperature patterns: the results obtained represent only a static photograph of a microclimate classification dependent only on land use and morphology of an urban area during a typical summer day (air temperature maximum around 30 °C and clear sky conditions). The aim is to give a climate classification of the entire urban area to urban planners to decide regeneration policies that take into account sustainability, mitigation and adaptation.

The urban structure (geometry characteristics) and the urban fabric (physical form) strongly influences the urban microclimate. Yang and Li [29] founded that the air temperature decreases to increase of H/W ratio, especially during the summer. Petralli et al. [35] report the results obtained for the city of Florence, which has a historical center similar to Bologna one, which is why it was chosen to use these parameterizations. 

In order to relate these classes of climatic morphology with a well-being index, which is defined as a condition of physiological equilibrium of the person who is in an outdoor environment subjected to the variability of atmospheric and metabolic parameters, this classification was then normalized through an up-scaling of the results of microclimate analysis. 

The five areas simulated with the Envi-met model were geo-localized using GIS software and, for each area, the maps of the PET values (physiological equivalent temperature) [43] obtained from the Envi-met fluid dynamics model were extracted during a heatwave (4 August 2017).

Through the intersection of the individual PET values obtained from Envi-met (with a resolution of 5 m) and the “blocks” of Bologna city, a PET value was obtained for each individual block (Figure 9). The upscaling was computed calculating the mean PET value of all blocks that belong to the same climate morphological class.

## 3. Results

The input map data, described in Section 2.2, applying the Equations (1)–(4), gives as a result the maps reported in Figure 10, Figure 11, Figure 12, Figure 13 where the temperature increases (or decreases) are reported as a function of the four considered variables. The obtained values were normalized from 0 to 1 in order to have comparable values between the four maps.

The map as a function of the surface temperature (Figure 10) shows the areas where an increase in the surface temperature is associated with an increase in air temperature, characterized by areas of orange/red. In Figure 11 the classes of air temperature were obtained from the input map of the vegetation: in this case, the relationship is inverse, that is: a large increase in air temperature corresponds to low values of the percentage of vegetation and in fact, the historical center appears all red (i.e., the maximum class).

Figure 12 shows the map obtained for the air temperature variations, normalized between 0 and 1, as a function of the building density: in this case it can be seen how the variability is very low, obtaining areas with blue color for almost the entire city of Bologna. The only areas in which the classes have high values are those of the historical center and other neighboring areas where the density of the building is significant.

Finally, the relationship with the H/W ratio (Figure 13) shows an inverse relationship as a function of the temperature variations: the areas where the increase in temperature is most significant are the hills and the periphery and not the historical center as was the case for the other three variables. This is due to the fact that this morphological index basically represents the free field of view and therefore takes into account the shadow that occurs in narrow canyons with adjacent tall buildings.

The urbanized territory of the city of Bologna was then catalogued according to classes with the same climatic morphology (MC) by carrying out a combination of the previously obtained classes of function between air temperature and the four input variables. Homogeneous MC means a portion of the territory falling within an interval belonging to the same category resulting from the combination of surface temperature, greenery and urban morphology.

The result of these climate classes is shown in Figure 14 where the value of the functional obtained was normalized between 0 and 1. It was decided to divide the index obtained into seven classes, and it can be seen how the historic center is characterized by the two major classes and how it differs from the classes obtained for the rest of the urban texture of the Bologna municipality. It is interesting to see how from a climate point of view the hilly area sometimes coincides with the valley areas characterized by vegetated land.

In order to relate these classes of climate morphology to a well-being index, the methodology explained in Section 2.3 and in Figure 9 was applied.

The PET values obtained for each “block” were normalized (between 0 and 1) and related to the climate morphology classes corresponding to the same “blocks” (Figure 15). The linear relationship shows how an increase in the MC classes corresponds to an increase in the average value of PET (increase in thermal discomfort): the analysis was carried out for two hours 10:00 and 14:00 in order to have PET values with a more significant variability. As regards the first class (0–0.3) and the last (0.85–1) shown in Figure 15, no “blocks” of the areas simulated by Envi-met corresponded to this classification, but the high value of correlation coefficient of the linear regression suggested that there is a clear linear trend. The standard deviation (σ) and the percentage of data that are in the ± σ of the mean value interval of all PET in the same block were computed and reported in Table 1 for the two hours 10:00 and 14:00. The greatest percentage of data are in the range mean PET ± σ.

Through these relationships between the equivalent physiological temperature and the MC classes, performing an up-scaling of the results of the micrometeorological modeling on the climatic morphology classes, the map of the normalized microclimate well-being (BMN index) was defined and computed for the whole Bologna “blocks” (Figure 16).

From the seven MC classes, we obtained a map with only four classes: the data showed the spatial and urban coherence, and we decided to aggregate them in order to give some replies to the local administration needs of effective management. For this purpose, we have to consider the aim for which the map was created: the map today describes the level of microclimate well-being, but in the near future it will potentially be used to indicate the percentage of improvement that the Municipality will expect from each portion of the city. Therefore, the map becomes a very useful general indication for the Urban Office which must evaluate the requests for direct design interventions and urban planning operational agreements. Therefore, an analysis of the urban texture was carried out which allows easier identification of the fragilities to help local administration in the urban regeneration interventions: the new classes obtained are extremely homogenous throughout the entire considered territory.

## 4. Discussion

There are many studies and modeling approaches to be able to fully define the urban microclimate, among the most widely applied is that of the local climate zones (LCZ) [44,45]. This approach solves the problem starting from a morphological classification of the urban fabric and has a wide planning and regulatory value for newly designed works, even if it has been applied, becoming urban guidelines, in various situations of consolidated cities [46,47,48]. The possibility of concrete application and transformation into urban planning rules is however mainly linked to the interaction between the application of the method and the database available in the municipalities that generally have few human resources available to carry out the transformation of their consolidated data with respect to application of this methodology.

The methodology presented in this study seeks to reconcile the presence of a consolidated data structure, characteristic of public administrations, with the possibility of a rapid application of the results to the city to provide a quick tool to be applied in the urban regulation. Although the physical processes that operate in the city are multiple and characterized by temporal evolutions on different scales (daily, seasonal), the objective of the method is to build a kind of photograph of the maximum impact within the urban fabric so that the regulations to follow can allow the non-exceeding of dangerous thresholds for urban well-being that would be reflected on the population for years. For these reasons, the study focuses on data relating to a heat wave that has affected the city in order to establish one of the worst cases for human well-being [49].

The methodology was developed on the basis of the request of the Municipality of Bologna to carry out a modeling study on five areas of the city that have a particular value in the protection of vulnerable groups and represent a large part of the urban-architectural characteristics of the city (Figure 4). The analysis was then carried out with the ENVI-met software initialized with the municipality database and the meteorological data available from the Dexter system. A rough correlation was therefore sought between urban structures and PET (Figure 15, Table 1) in order to be able to perform a scaling-up to the rest of the city, bearing in mind that the particularity of the situation of the city center, of medieval structure, would have represented a marked criticality in the analysis conducted with this modeling, and for which further specific research is indispensable. The central part of the city which consists of large squares but also of a strong built structure is the least flexible part of an urban regeneration. What can certainly be assumed for this part of the city, poorly equipped with green infrastructures, is that in a radial structure composed of narrow urban canyons during the day part of the city will be exposed to shade-sunshine conditions depending on the solar angle, with a consequent strong differentiation of the impacts on well-being, while at night, due to radiative exchange between buildings, the conditions of well-being for the weaker groups will tend to worsen ubiquitously [50,51]. Thus, the city center, even if not completely resolved by the study, provides a clear indication in the dialectic between municipalities and operators in the construction sector: if we want to intervene, given the limitations determined by the particular built environment, urban regenerative forms must include reduction solutions of the thermal load of this part of the city, and in so doing it is clearly oriented towards solutions related to green roofs, and to the optical properties of surface materials, unfortunately the green walls can only represent few opportunities for application in this context.

The availability of satellite images and an excellent census of urban greenery (Figure 5 and Figure 6), together with the data available at the municipality, have therefore made it possible to solve the system of four equations on which this study is based and to arrive at a microclimatic characterization of the city with the specific purpose of directing regeneration policies not only with a generic built approach, but which also contained a strong climatic indication as also provided for by the Covenant of Majors [6].

## 5. Conclusions

The study highlighted the possibility of indexing the city structure in terms of “blocks” for the definition of levels of well-being through a micro-climate approach. The model obtained is very realistic as it respects the morphological and structural characteristics of the urban composition present in the city. However, further steps will be necessary for the optimization of the BMN index values. The parameterizations utilized, especially the relationship between air and surface temperatures, although clearly consistent, must be implemented with greater precision by means of other studies to be conducted on the territory. The results obtained are, however, to be considered ready for operational use since potential differences can have, even minimal, influence on the results on specific characteristics of the urban morphology and microclimate.

The present study’s findings will become part of the PUG as a benchmark to be utilized when private, and public actors want to regenerate part of the city. This undoubtedly represents a step forward in terms of a real climate-centered approach considering the impressive evidence of the built environment’s role on the wellness of the resident population [52,53,54].

As mentioned, the analysis showed a marked consistency with the areas of the structured territory and to be structured, also identifying the particular weaknesses in each area. It should be emphasized that the old town, although homogeneous with respect to the general representation, needs particular attention as the scale characteristics of its specific morphology do not perfectly match that of the model and specific in-depth studies, including modeling must support any interventions.

## Figures and Tables

**Figure 1 ijerph-18-04898-f001:**
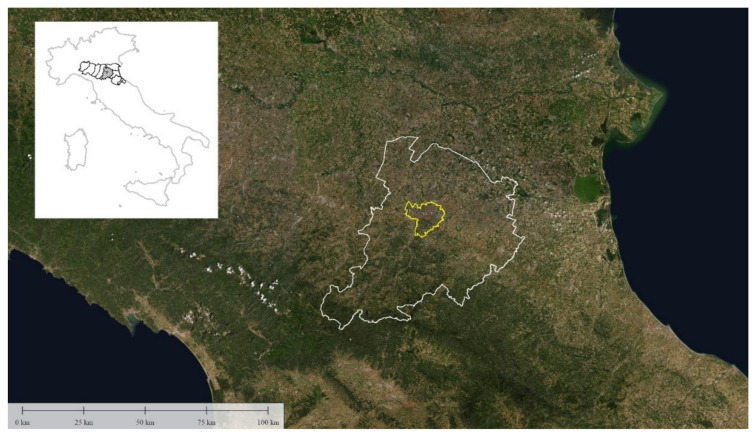
Borders of Bologna Metropolitan city (white) and of Bologna Municipality (yellow) on a satellite image.

**Figure 2 ijerph-18-04898-f002:**
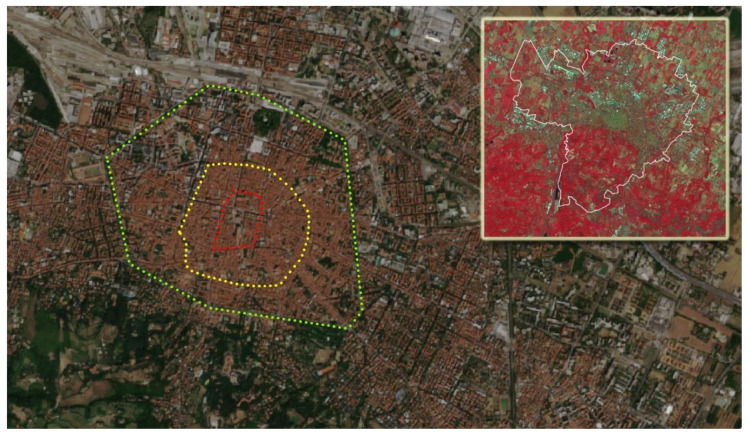
The historical center of Bologna with the three sets of ancient walls (today almost completely destroyed), signing successive development phases of the city: the first set (“selenite circle”, in red) built between the 4th and 10th centuries, the second (“Cerchia del Mille”, in yellow) constructed around the year 1000, the third built between 1327 and 1390 (in green). In the insert: false color Sentinel-2 satellite image for a larger area, highlighting in red the vegetated areas.

**Figure 3 ijerph-18-04898-f003:**
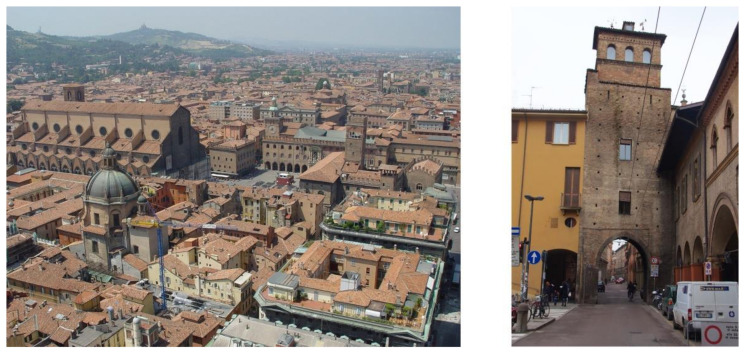
(**Left**): oblique view of the city center [25]. (**Right**): Torresotto in Via San Vitale [26], where it is possible to see on both sides the traditional arcades of Bologna currently considered as a World Heritage Site by UNESCO.

**Figure 4 ijerph-18-04898-f004:**
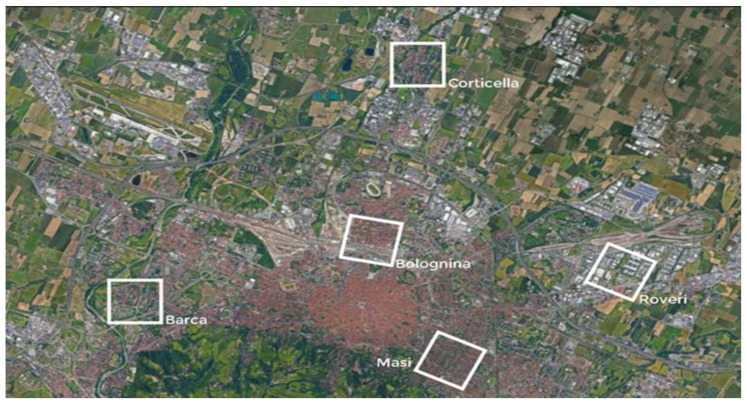
Geolocation of the five study areas of the Bologna municipality simulated with the Envi-met model.

**Figure 5 ijerph-18-04898-f005:**
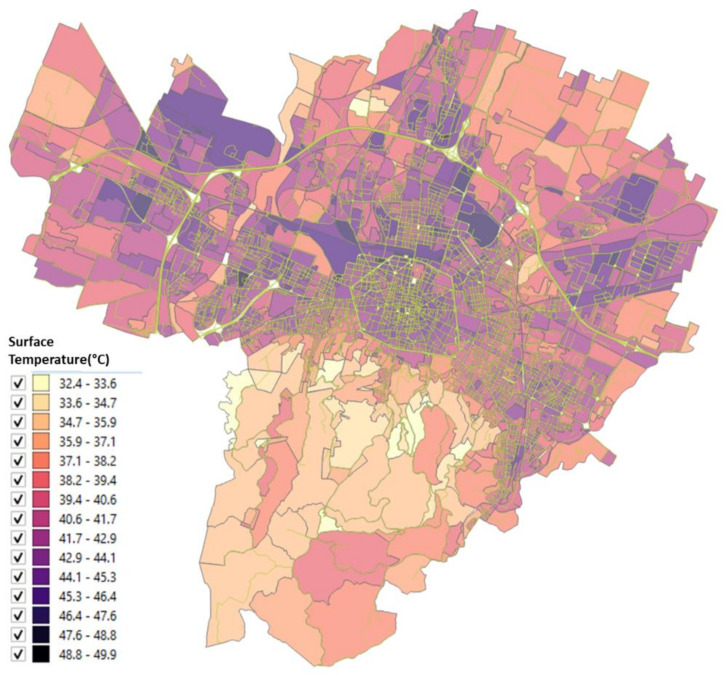
Surface temperature from Aster sensor data for each block of the Bologna Municipality (23 June 2017 at 12:10 a.m.).

**Figure 6 ijerph-18-04898-f006:**
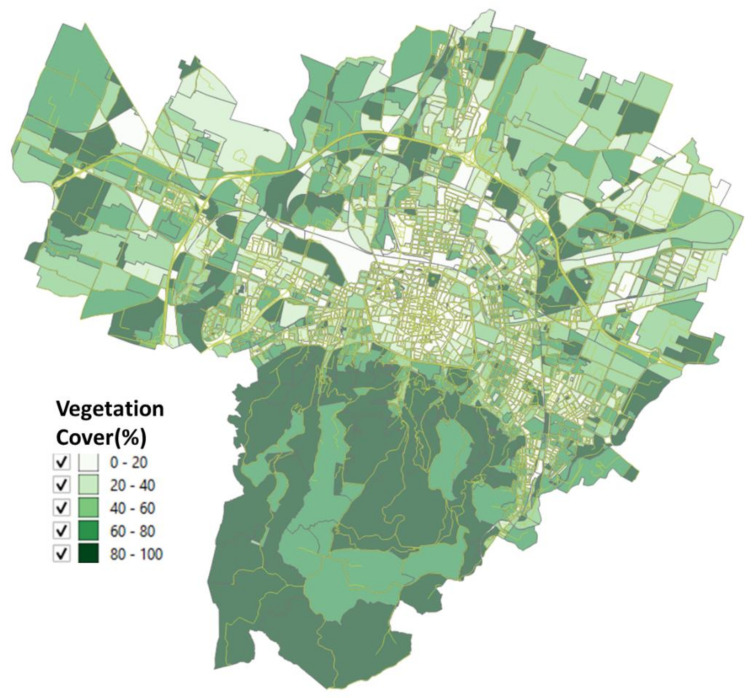
Percentage of vegetation cover for each block of the Bologna Municipality derived from Sentinel-2 satellite data.

**Figure 7 ijerph-18-04898-f007:**
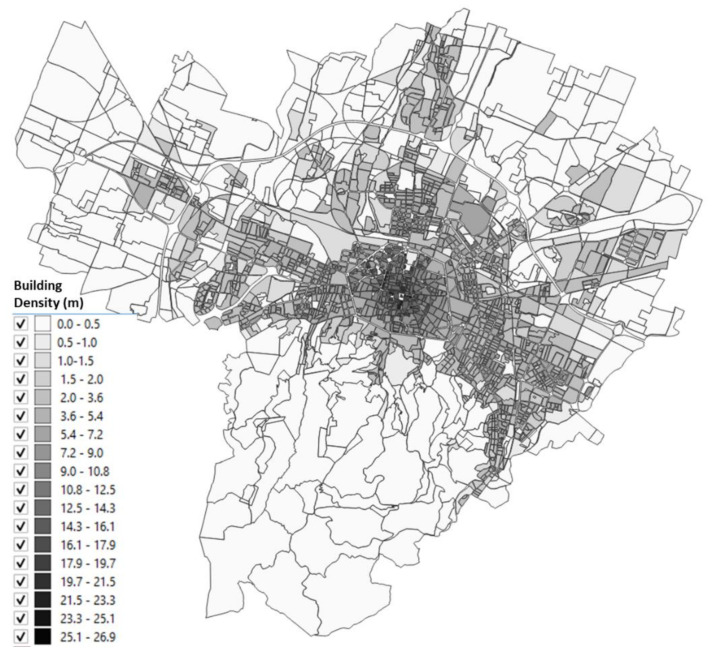
Building density for each block of the Bologna Municipality.

**Figure 8 ijerph-18-04898-f008:**
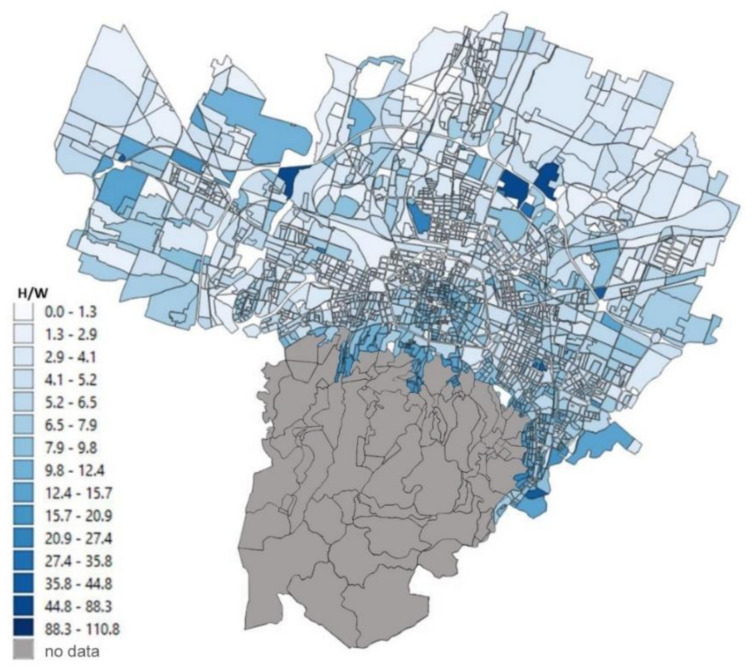
H/W ratio (building height/road width) for each block of the Bologna Municipality.

**Figure 9 ijerph-18-04898-f009:**
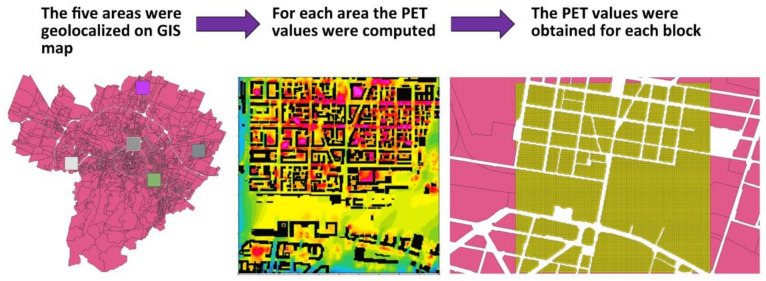
Geo-localization on GIS map of the five simulated areas, extraction of PET values from the model runs and computation of PET values for each “block”.

**Figure 10 ijerph-18-04898-f010:**
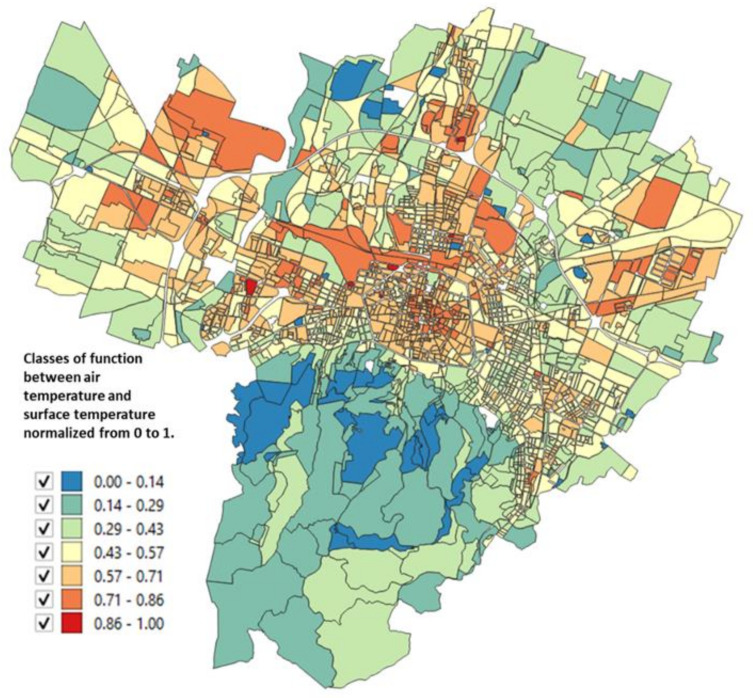
Relationship between the air temperature and the surface temperature normalized from 0 to 1 obtained from Unger et al. [34] (based on Equation (1)).

**Figure 11 ijerph-18-04898-f011:**
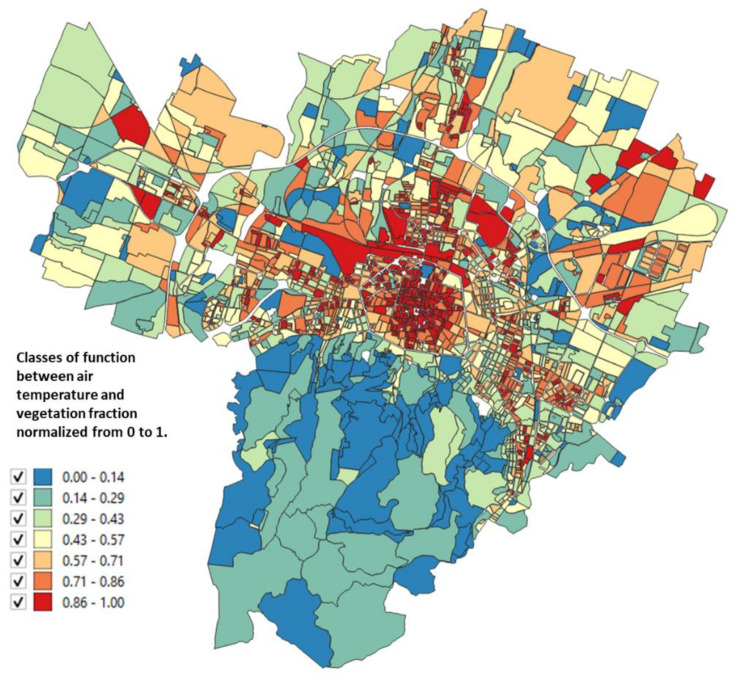
Relationship between the air temperature and the vegetation fraction normalized from 0 to 1 obtained from Petralli et al. [35].

**Figure 12 ijerph-18-04898-f012:**
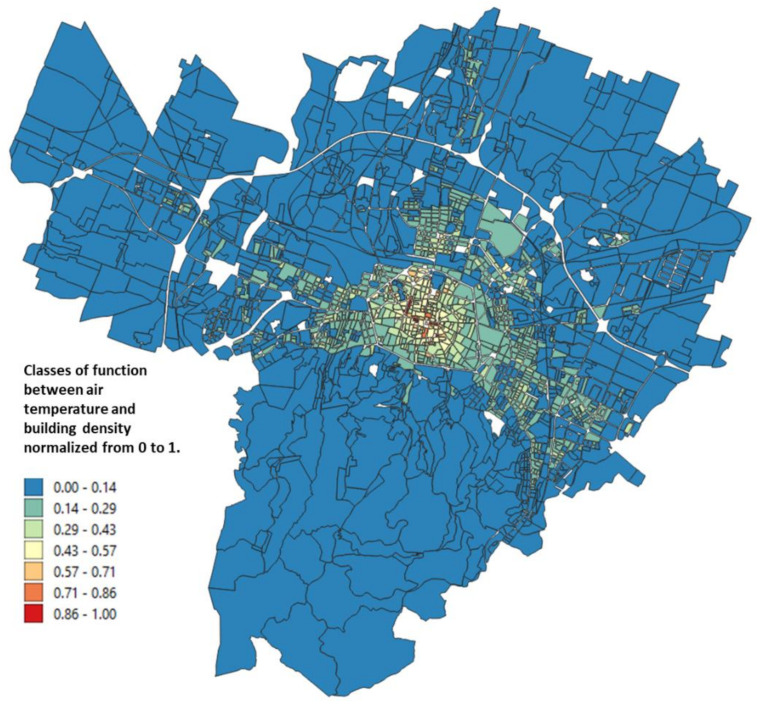
Relationship between the air temperature and the building density normalized from 0 to 1 obtained from Petralli et al. [35].

**Figure 13 ijerph-18-04898-f013:**
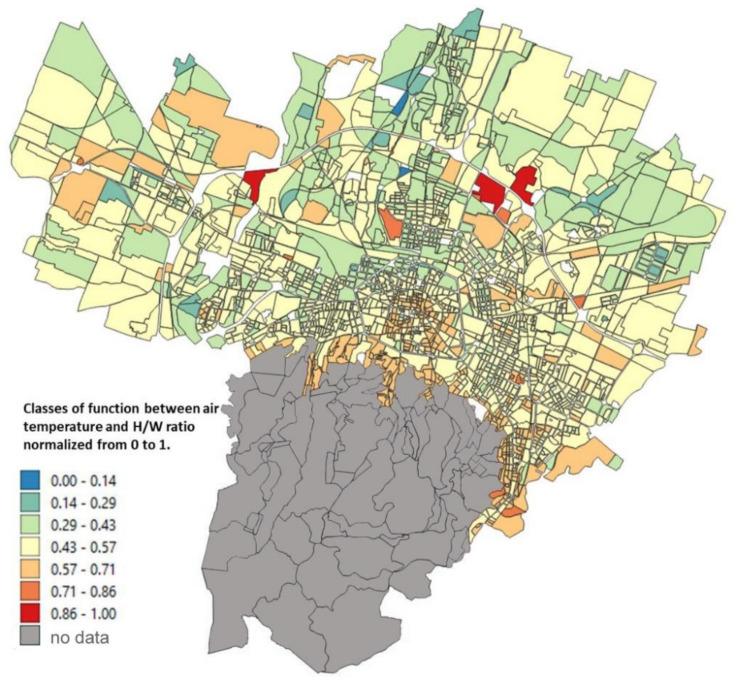
Relationship between the air temperature and the H/W ratio normalized from 0 to 1 obtained from Oke [36].

**Figure 14 ijerph-18-04898-f014:**
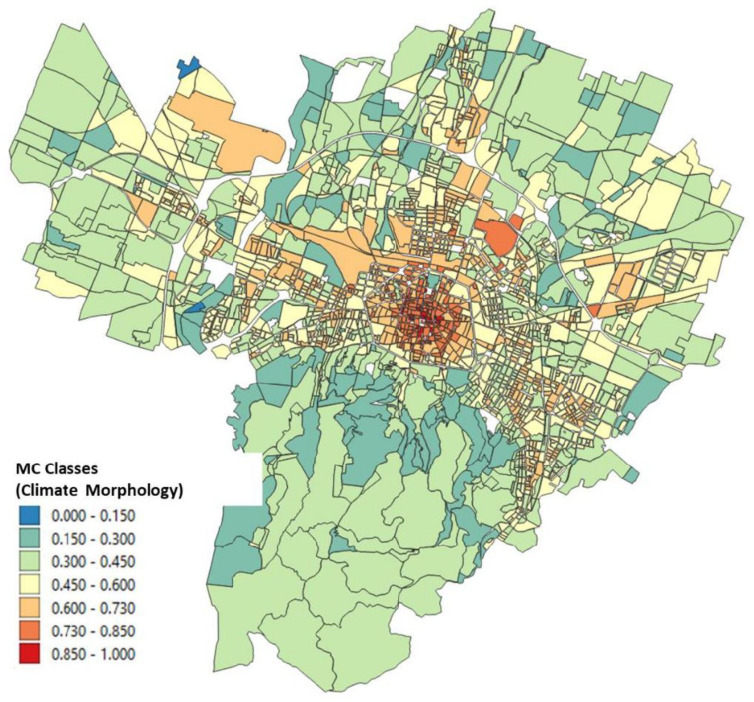
Climate morphology classes normalized between 0 and 1.

**Figure 15 ijerph-18-04898-f015:**
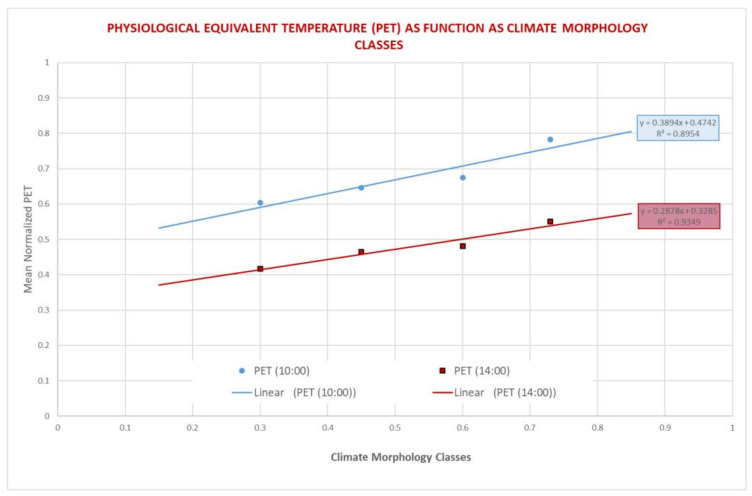
PET values, normalized between 0 and 1, as function of climate morphology classes (MC) during 10:00 a.m. and 14:00 p.m. and the relative linear regressions.

**Figure 16 ijerph-18-04898-f016:**
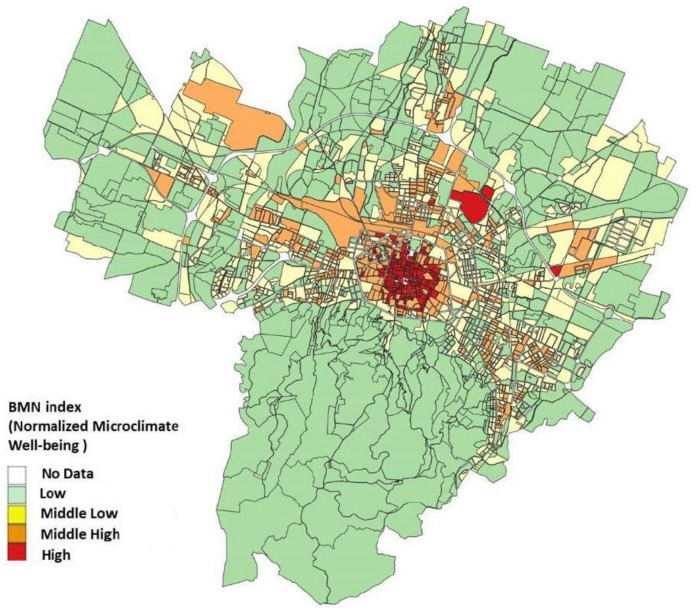
Normalized microclimate well-being index with relative sensations of thermal comfort.

**Table 1 ijerph-18-04898-t001:** Mean PET values, normalized between 0 and 1, for each climate morphology classes (MC) during 14:00 a.m. and 10:00 p.m. and the relative standard deviation and the percentage of cases that are in the interval mean PET ± σ.

h 14:00			
**MC CLASSES**	**MEAN PET**	**DEV. ST. PET**	**%Cases ± σ**
**0.15–0.30**	0.60	0.24	57
**0.30–0.45**	0.64	0.21	66
**0.45–0.60**	0.67	0.18	68
**0.73–0.85**	0.78	0.08	92
**h 10:00**			
**MC CLASSES**	**MEAN PET**	**DEV. ST. PET**	**%Cases ± σ**
**0.15–0.30**	0.42	0.22	58
**0.30–0.45**	0.46	0.22	58
**0.45–0.60**	0.48	0.21	64
**0.73–0.85**	0.55	0.23	58

## Data Availability

Data sharing not applicable.

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
