# Peer review of "Microclimate Classification of Bologna (Italy) as a Support Tool for Urban Services and Regeneration"

_ijerph, 2021, doi:10.3390/ijerph18094898_

Round 1

Reviewer 1 Report

The submitted manuscript is a study that presents a methodology to characterize the spatial variation of thermal comfort in Bologna on a building block scale. While the approach is interesting and I like the idea to upscale Envi-met simulations, the manuscript has some shortcomings that I address in more detail below. Major points are that a Discussion Section is currently completely missing, that H/W ratios depicted in Fig. 13 are artificially inflated and that – in my view - a linear correlation between essentially air temperature and PET (at daytime) is problematic.

For these reasons and further detailed comments given below, my recommendation is to reject the manuscript from publication in IJERPH. 

Remarks:
A discussion is completely missing. A comparison with earlier approaches should be conducted, and reasoning should be provided why the approach described in the study was favored against e.g. the local climate zone approach.
It is convention to have a space between numbers and units.

Materials and Methods:

The chosen study areas need to be described in more detail. In what aspects do these areas differ from each other and are these areas in total representative for Bologna? I think it is not, because the old town is not represented, which is crucial for the conclusions of the study.

Results:
I think that the H/W values given for the southern, vegetated areas (of up to 110) are unrealistically high. This translates e.g. even for a very narrow road of 5 m to a mean building height of 550 m…this is problematic since it is used in equation 4 to calculate a UHI effect. You do not have a UHI effect in hilly, wooded terrain.
In Figure 15 you used only 4 points for the regression, but you describe that you had 5 study areas with Envi-met. That means one point is missing here. Besides that, I think the conclusions drawn from this regression are problematic, since air temperature is only one of several meteorological variables that control PET. PET is very dependent on global radiation. That means narrow streets (which you have in old town), which provide shade generally lead to low daytime PET values. Now, the results of this study suggest the opposite because of this assumed linear relationship. 
To solve this I think it would be better to look at daytime PET, but also at UHI. What you might have is low daytime PET in old town, but also a pronounced UHI during evening/ night-time because of the narrow streets which lead to longwave radiation trapping. So in the end, you might be right that the overall thermal comfort in old town is not the best overall, but currently – in my view – for the wrong reasons.

Detailed remarks:

l. 28: do you mean “in the last 30 years”?
l. 41: I suggest changing “only mitigation” to “only climate change mitigation”

l. 43: “rethink to the” -> “rethink the”

l. 70-71: I’m not sure what this sentence means and if “declined” is the right word here.

l.84: maybe “urban expansion”?

l.100-106: what kind of impact does this have on the urban planning process of Bologna?

l. 162-163: rather “exchange processes of heat and vapour between ground, wall and vegetation surfaces and the atmosphere”

l. 164: Tsoka et al. (2018) could be added here. 

l. 166: initial and boundary conditions

l. 176-177: what are the reasons for a high/low vulnerability assessment?

l. 183: “the formal model used badly” – I’m not sure what this means, maybe a translation issue

l. 188: This is horizontal resolution I presume. What was the vertical resolution? What was the height of the model domain?

l. 197: not only initialization, but also boundary conditions

l. 199: “Nord” -> “North”

l. 203: typo

l. 205-206: are there references for this methodology or is this a new methodology? Why was this method chosen over e.g. local climate zone classification?

l. 208-211: what was the time and date?

Fig.5: please add time and date when the data was recorded.

l. 230-231: this definition needs to be in point 3

l. 232-235: this definition needs to be in point 4

l. 239-240: where is this index shown? I think the correct term is “normalized difference vegetation index”
l. 255: “surface temperature” – which unit?

l. 256: “HW” – this is not necessary, since you already defined H/W ratio

l. 270-271: this seems to contradict eq. 4.

l. 277-278: “was the normalized through an up-scaling of the results of microclimate analysis” -  Explain what you mean by this in more detail please. I read between the lines that you calculated a temperature value based on equations 1-4 and then you related this (how?) to the calculated PET with Envi-met. Please clarify.

l. 284: do you mean you calculated the average PET for each block? A metric for the variability in each block could be interesting as well.

l. 301 Please clarify that this is the calculated air temperature based on eq. 1.

l. 311-316: this does not make sense. High H/W means high buildings and/or narrow streets. This is obviously not the case for the southern, highly vegetated areas. You can also see from the measured surface temperature data that it must be cooler in the southern periphery and hills.

l. 324: what do you mean by “linear combination”? I assume you summed the normalized values and the normalized again by the maximum value? 

Fig. 16: I would suggest changing the wording from “fragility” to “heat exposure”

l. 387: here, literature that connects thermal comfort to health issues would be a better fit.

References:
Tsoka, S., Tsikaloudaki, A., & Theodosiou, T. (2018). Analyzing the ENVI-met microclimate model’s performance and assessing cool materials and urban vegetation applications–A review. Sustainable cities and society, 43, 55-76.

Reviewer 2 Report

This study presents an important analyses for the local administration of the resident population. While I found this manuscript well-developed, the introduction part seems a bit far from the main goal of this study. The part could be more tightened to present what is the main goal of this study and why this study is important and valuable.
